# Potential of Calabash (*Lagenaria siceraria*) and Sweet Potato (*Solanum tuberosum*) for the Remediation of Dichlorodiphenyltrichloroethane-Contaminated Soils in Tanzania

**Hamisi J. Tindwa [1] and Bal Ram Singh [2,\*]**

[1] Department of Soil and Geological Sciences, College of Agriculture, Sokoine University of Agriculture, Morogoro P.O. Box 3008, Tanzania; tindwa.hamis@sua.ac.tz

[2] Faculty of Environmental Sciences and Natural Resource Management, Norwegian University of Life Sciences, P.O. Box 5003, 1433 Ås, Norway

\* Correspondence: balram.singh@nmbu.no

**Abstract:** A study was conducted to test the potential of calabash, sweet potato, pumpkin, simsim and finger millet to phytoaccumulate dichlorodiphenyltrichloroethane (DDT) and its metabolites from NHC Morogoro- and PPO Tengeru-contaminated sites. Parallel field and screenhouse-potted soil experiments were performed to assess the efficacy with which the test plants phytoaccumulate DDT from the soil. In the screenhouse experiment, treatments were laid out following a split-plot arrangement in a completely randomized design (CRD), with the main plots comprising two DDT concentration levels–low (417 mg kg$^{-1}$) or high (2308 mg kg$^{-1}$)—and the plant species *Cucurbita pepo*, *Lagenaria siceraria*, *Ipomoea batatus*, *Sesamum indicum* and *Eleusine coracana* were considered as subplots. A field experiment with the same crop species as the treatments was laid out in a randomized complete block design, and both experiments were performed in triplicate. In addition to determining the concentration of persistent organic pesticides in the soil profile, parameters such as the total DDT uptake by plants, shoot weight and shoot height were monitored in both potted soil and open field experiments. Overall, calabash and sweet potato exhibited the highest (4.63 mg kg$^{-1}$) and second highest (3.45 mg kg$^{-1}$) DDT concentrations from the high residual DDT potted soil experiment. A similar trend was observed when the two plants were grown in low DDT soil. Sweet potato recorded the highest shoot height and weight in the potted soil experiments, indicating that increasing amounts of DDT had a minimal effect on the plant's growth. Although sweet potato outperformed calabash in the amounts of DDT concentration in the shoots under open field experiments, the uptake of DDT by calabash was the second highest. Calabash—a wild non-edible plant in Tanzania—presents a potential phytoremediation alternative to edible and much studied pumpkin.

**Keywords:** soil remediation; contaminated soils; phytoaccumulation; persistent organic pesticides; Tanzania

## 1. Introduction

The remediation and management of soils, waters and sediments contaminated with persistent organic pesticides (POPs) has become a global environmental priority, mainly because POPs are hazardous to life. POPs, originally manufactured for pest and disease control, crop production and other industrial processes, have a high degree of persistency in the environment, potential for long-range transportability, the ability to bioaccumulate in the lipid components of living systems and a high toxicity, even at very low concentrations [1,2]. Considering these problems associated with the use of POPs, the latter were banned worldwide [3], and research into effective clean-up methods evolved over time. Most of the developed world has intensive clean-up programs for POP-contaminated sites

by developing suitable and effective methodologies [4]; however, the research on such methodologies, especially on phytoremediation in SSA countries, is still rather scanty.

All clean-up methods currently in use can be grouped into three main categories. Category one is a collection of methods based on the principle of containment in which the contamination is sealed in a protective barrier to limit its release, and it may include practices such as land filling [5] and solidification and stabilization practices [6]. Category two involves a collection of technologies designed to destroy the POPs either through (a) non-combustion methods, such as dehalogenation, or (b) combustion methods like incineration or thermal desorption, which break down POPs to simple compounds such as $CO_2$, methane ($CH_4$) and water ($H_2O$) [7–9]. Category three, on the other hand, includes a collection of technologies that involve the extraction of the contaminant from the matrix (soil, sediment or water) through either (i) concentration or (ii) the liberation/stripping of the contaminant, paving way to the treatment of the liberated contaminant through convenient methods of category one or two above. Examples of methods in category three include ex situ soil washing, ex situ solvent extraction, in situ soil flushing, soil vapor extraction, ex situ bioremediation and in situ bioremediation [10,11].

The deployment of any one or a combination of the methods mentioned above for the remediation of contaminated sites can be influenced by considerations for technical, economical and operational feasibility. As a rule of thumb, high technology demanding, expensive, time-sensitive and highly efficient technologies such as incineration can be feasible in the developed world, while time consuming, low-input yet efficient technologies such as bio- and phytoremediation are both suitable and feasible in most of the developing world.

The use of higher plants as detoxifiers, filters or traps of POPs and other pollutants is a well-proven approach, but this approach, also known as phytoremediation, is slow and difficult due to the inherently low bioavailability of POPs to plants [12,13]. A handful of research efforts have reported the successful testing of various higher plants such as *Cucurbita pepo* and *Cichorium intybus* as potential tools for the phytoremediation of POPs, including DDT- and metabolite-contaminated sites [14,15].

The success of phytoremediation depends mainly on the properly selected plant species [16], which show fast growth, large biomass production in a short time, a developed root system, a higher tolerance to pollutants and the ability to accumulate toxins in aboveground parts, and resistance to diseases, pests and weather conditions. The test plants used in this study meet all or most of these properties. Furthermore, the uptake and accumulation of pollutants in crop plants vary with species or cultivars, the typology of pollutants as well as the level of contamination [17].

In some cases, edible crop species can exhibit all or most of the above properties of a good phytoaccumulator plant. Such plants can be used as agents of phytoremediation efforts if the produced phytobiomass is protected from animal and human consumption to avoid their movement up the food chain [18]. This study was designed, therefore, to test the potential of calabash—a locally available non-edible plant—along with simsim, pumpkin, sweet potato and finger millet to phytoaccumulate DDT from contaminated soils. The results would inform both the health risks associated with growing the edible species on the contaminated soils as well as their potential for use in the phytoremediation of contaminated soils.

## 2. Materials and Methods

### 2.1. Description of Study Sites

The Plant Protection Office (PPO) at the Tengeru site is in Arusha, Tanzania at a longitude and latitude of 3.39092 and 36.799200. Records show that the PPO Tengeru site was once used as a locust control center for Tanzania and thus had tons of supplies of pesticides, including malathion and lindane, which are stored there [19]. In 2000, pesticides packed in drums and stored in open air started rusting and were subsequently buried in soil at the PPO Tengeru backyard (Scheme 1).

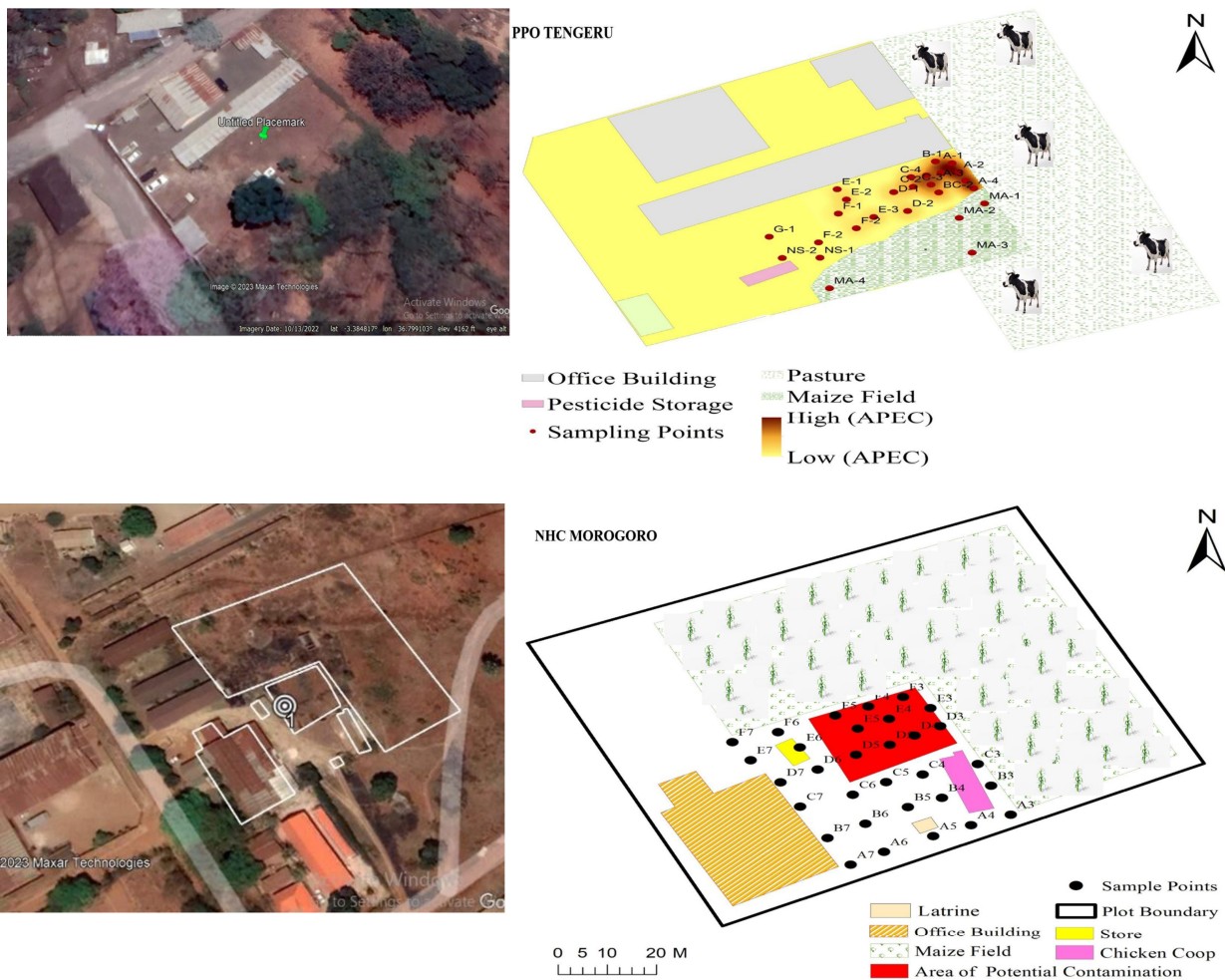

**Scheme 1.** Google Maps extract showing location of PPO Tengeru study site (**above**) and NHC Morogoro study site (**below**).

NHC Morogoro study site, on the other hand, is located at a latitude and longitude of 6.083333 and 37.666667 in the heart of the Morogoro Municipality (Scheme 1). The site was originally used as a center for the formulation and repackaging of DDT and endosulfan, among other pesticides, for onward distribution to end users across the country [3]. Following an international ban on the production and use of these chemicals due to perceived side effects to humans and the environment, activities were stopped, and the site was eventually abandoned in 1997 [3].

*2.2. Soil Sampling and Analysis*

Soil sampling at the NHC Morogoro and PPO Tengeru sites was carried out for two main purposes, namely for soil fertility and physicochemical analysis, on one hand, and to estimate the level of contamination by POPs, on the other hand. Soil samples for physicochemical and soil fertility analysis were taken using an auger to a depth of 30 cm from the soil surface. The hand auger used for surface soil sampling was rinsed with distilled water after each sampling. Samples were transferred to the pre-cleaned amber bottles. Each soil sample was 500 g, enough for specific analysis and a control sample. Samples were transported to the laboratory at Sokoine University of Agriculture for analysis. Soil fertility analysis parameters included macronutrients N, P, K, S, Ca and Mg and micronutrients Mn, Fe, Cu and Zn as well as CEC, EC and OC. All of these nutrients were determined following the procedure described in [20]. The physical and chemical characteristics of the soils at the study sites are as shown in Table 1.

**Table 1.** Physical and chemical properties of the soil.

| Soil Property | Value and Rating | |
|---|---|---|
| | **NHC Morogoro** | **PPO Tengeru** |
| pH 1.2.5 $H_2O$ | 7.02 n | 6.78 n |
| OC% | 2.36 m | 2.67 n |
| Total N% | 0.53 h | 0.67 h |
| OLSEN available P (mg kg$^{-1}$) | 2.90 l | |
| Bray 1 Available P (mg kg$^{-1}$) | | 2.19 l |
| CEC cmol(+)kg$^{-1}$ | 11.40 l | 10.9 l |
| Na cmol(+)kg$^{-1}$ | 0.15 l | 1.11 l |
| K+ cmol(+)kg$^{-1}$ | 2.51 h | 2.71 h |
| Mg++ cmol(+)kg$^{-1}$ | 2.95 m | 2.90 m |
| Ca++ cmol(+)kg$^{-1}$ | 3.53 h | 3.83 h |
| Sand% | 56.20 | 28 |
| Silt% | 9.30 | 35 |
| clay% | 32.50 | 37 |
| Textural class | Sandy clay loam | Clay loam |

The ratings were according to Landon (1991). n = neutral; m = medium; h = high; l = low.

Separately, surface soil samples meant for POP testing were put in pre-cleaned amber bottles and immediately placed in cool boxes with ice packs to be transported at the end of the sampling day to the laboratory for analysis. Special drilling equipment hired from the water solutions drilling company (WSDC) were used for sample collection from up to seven meters below the soil surface at the PPO Tengeru site. At the NHC Morogoro site, samples up to 2 m below the soil surface were taken using open ditch profile openings. The soil samples from both sites were placed in cleaned amber bottles and transported to two separate laboratories, namely the Tanzania Plant Health and Pesticides Authority (TPHPA) and Tshwane University of Technology's environmental and water chemistry laboratory in South Africa. At both laboratories, soil samples were analyzed for DDT, lindane, fenthion, diazinon and permethrin.

Soil samples (2.5 g) were weighed in pre-cleaned cellulose thimbles and were thereafter transferred into cleaned Soxhlet apparatus for extraction. The samples were extracted for 16 h using a mixture of n-hexane/acetone (2:1, *v/v*). Upon completion, the extracts were allowed to cool down to room temperature. The extracts were carefully transferred into pre-cleaned round-bottom flasks and were rotary-evaporated to approximately 2 mL. Two laboratory reference soil samples were spiked with known amounts of the targeted compounds, and they were similarly prepared following the procedures named above. Spiking the reference samples serves as a quality assurance measure to assess the efficiency of the extraction method by estimating the analytical recoveries of the targeted compounds.

The soil extracts were purified using deactivated silica gel packed into glass columns. Prior to the clean-up procedure, the packed columns were conditioned using 25 mL of n-hexane to remove trapped air and possible interfering contaminants. The concentrated extract was quantitatively transferred into the glass column and was eluted under gravity with 40 mL of n-hexane/acetone (2:1, *v/v*). The eluate was rotary-evaporated to approximately 2 mL and transferred into an amber vial, where it was further concentrated with high-purity nitrogen gas until it reached incipient dryness. The extracts were re-constituted with 1 mL of Toluene, followed by the addition of 50 µL of 500 pg µL$^{-1}$ of DDT-d8 that was employed as an internal standard.

Quantitative estimation of all of the targeted compounds was performed using an Ultra-trace 2010 Shimadzu GC equipped with QP 2010 Ultra mass spectrometer (Shimadzu, Kyoto, Japan) operated in EI mode. The chromatographic separation of these compounds was achieved using DB-5 MS (15 m, 0.25 mm i.d., 0.10 µm film thickness) capillary column. The optimal conditions employed for the GC-EI-MS instrument are shown in Table 1. To enhance the sensitivity of the instrument and to overcome the inherent problems of interfering co-extractants, the mass spectrometer (MS) acquisition was carried out in selected

ion monitoring (SIM) mode. In this case, a target ion and two reference ions were selected for each targeted compound as well as internal standard (DDT-d8) for their identification and quantification.

### 2.3. Pot and Field Experiments

Parallel field and screenhouse potted soil experiments were performed to assess the efficacy of phytoaccumulator test plants for DDT from the soil. The potted soil screenhouse experiment was laid out with a split-plot arrangement in a completely randomized design (CRD) with three replications. The main plots comprised two levels of DDT, i.e., a low level of DDT concentration (total DDT = 417 mg kg$^{-1}$) and a high level of DDT concentration (total DDT = 2308 mg kg$^{-1}$). The two levels used in this study were chosen randomly to reflect actual concentration scenarios in contaminated sites available in Tanzania [19]. The test crops, namely pumpkin (*Cucurbita pepo*), calabash (*Lagenaria siceraria*), sweet potato (*Ipomoea batatus*), simsim (*Sesamum indicum*) and finger millet (*Eleusine coracana*), were considered as subplots. Plants were regularly watered (every other day) each time the soil moisture was raised to about field capacity. About 250 mL of irrigation water per pot was enough to bring the moisture status to field capacity. Three weeks after planting, pre-calculated amounts of all deficient essential plant nutrients were added to recommended quantities to support the growth and development of test plants by mixing them with irrigation water.

The field experiments were laid out in a randomized complete block design (RCBD) with three replications. The sloping terrain was used as a blocking factor, and, like in the potted soil experiment, the test crops were pumpkin (*Cucurbita pepo*), calabash (*Lagenaria siceraria*), sweet potato (*Ipomoea batatus*), simsim (*Sesamum indicum*) and finger millet (*Eleusine coracana*). Plants were established through direct seeding on 2 × 4 m plots. All deficient nutrient elements (Table 1) were corrected to recommended levels using mineral fertilizers.

### 2.4. Plant Sampling and Analysis

Plant sampling in the pot and in the field experiments was carried out at flowering, roughly 60 days after planting. Plants sampled included shoots (above-ground biomass) and roots (below-ground biomass). For the potted soil experiment, shoots were harvested by cutting at about 1 cm above the soil, and then soil in the pots was poured on to a manila sheet, and the roots were collected/removed by hand and washed with tap water and then rinsed in distilled water. The samples (shoots and roots) were kept in separate amber bottles, clearly labeled and transported to TPHPA laboratory for analysis of DDT, its metabolites and other pesticides. A duplicate set of samples was sent to the Tshwane University of Technology's environmental and water chemistry laboratory in South Africa for analysis. Prior to analysis in the laboratory, fresh weights of roots and shoots were measured. After weighing, shoots and roots were dried at about 50 °C for one week before being finely ground for chemical analysis.

The root and shoot parts of all test plants were prepared for the determination of residual levels of the targeted persistent organic pollutants. Approximately 2.5 g of the plant samples was weighed into pre-cleaned amber bottles. The samples were soaked overnight with 50 mL of n-hexane/acetone (1:1, *v/v*), followed by ultrasonic-assisted extraction for 30 min. The set-up was allowed to cool down to room temperature, and the extract was carefully transferred into a clean round-bottomed flask. The extraction was repeated using the same volume of extraction solvent and time. The extracts were combined and subjected to rotary evaporation as was previously indicated for soil samples. Two laboratory reference plant samples (lettuce) were spiked with known amounts of the targeted compounds, and they were similarly prepared following the aforementioned procedures. The spiked reference samples were employed for the estimation of the analytical recoveries of the target compounds. The plant extracts were purified using deactivated silica gel packed into glass columns as explained for soil samples, and quantitative estimation followed similar

procedure as that described for soil samples. Optimized conditions for the GC-EI-MS employed for the analysis of target compounds are summarized in Table 2.

**Table 2.** Optimized conditions for the GC-EI-MS employed for the analysis of target compounds.

| Parameters | Optimum Conditions |
| --- | --- |
| GC parameters | |
| Injection volume | 1 μL |
| Carrier gas (% purity) | Helium (99.999%) |
| Injection mode | Splitless |
| Flow control mode | Linear velocity |
| Injector temperature | 270 °C |
| Linear velocity | 63.5 cm/s |
| Column flow | 1.5 mL/min |
| Purge flow | 3.0 mL/min |
| Equilibrium time | 3.0 min |
| Sampling time | 2.00 min |
| Oven temperature programming | 70 °C held for 1.0 min, ramped @ 25 °C/min to 180 °C, ramped @ 8 °C/min to 300 °C and held for 5 min |
| MS parameters | |
| Ion source temperature | 270 °C |
| Interface temperature | 280 °C |
| Solvent cut time | 2.0 min |
| Acquisition mode | SIM |
| Ionization method | EI |

*2.5. Disposal of Polluted Soils and Plants*

All polluted materials (remaining soil samples and harvested plants) generated from this study were disposed through a government-certified agent following standard procedure provided by government chemist laboratories.

*2.6. Statistical Analysis*

The data on concentrations of DDT, its metabolites and other pesticides were subjected to Analysis of Variance (ANOVA) using the GenStat (14th edition) statistical software. Mean separation was carried out according to Duncan's New Multiple Range Test (NMRT) at $p = 0.05$ significance level.

**3. Results and Discussion**

*3.1. POP Concentration in Soils at PPO Tengeru and NHC Morogoro*

Several soil samples from the PPO Tengeru site contained DDT, lindane fenthion and diazinon in levels well above the permissible limits (Table 3). The total DDT concentration in the soils at Tengeru ranged from as low as 0.005 to as high as 11.36 mg kg$^{-1}$ of soil. All common isomeric forms of DDT and its metabolites, namely pp-DDT, op-DDT, pp DDE, op-DDE, pp-DDD, op-DDD and op-TDE, were detectable in samples from the PPO-Tenguru site. Similarly, α-lindane, β-lindane and γ-lindane were detected. At one sampling point, the total lindane amount recorded was 12.94 mg kg$^{-1}$, which is several folds above the permissible levels of 2.0 mg kg$^{-1}$ of soil in Tanzania. Other POPs that were detected at levels above the permissible limits at the Tengeru site included fenthion and diazinon (Table 3).

DDT was the dominant contaminant, as seen in the soil profile, and values above the international permissible limits (Canadian limits) were detected within 0.5 to 2.0 m below the soil surface in four out of five soil profiles opened at PPO Tengeru (Table 4).

**Table 3.** Pesticide concentration in surface soil samples at PPO Tengeru site, Arusha, Tanzania. Values above the permissible levels are marked in red.

| Field Sample ID | A2 | A3 | A4 | B1 | B4 | BC2 | C3 | E2 | F2 | NS1 | NS2 | TZ | Canada |
|---|---|---|---|---|---|---|---|---|---|---|---|---|---|
| **Pesticide** | | | | | | Concentration (mg kg$^{-1}$ Soil) | | | | | | \multicolumn{2}{c}{Permissible Level (mg kg$^{-1}$)} | |
| Total Lindane | | | 12.940 | | | | | | | | | 2.000 | 0.01 |
| α-Lindane | <0.056 | <0.056 | 5.400 | <0.056 | <0.056 | <0.056 | <0.056 | <0.056 | <0.056 | <0.056 | <0.056 | 2.000 | 0.01 |
| β-Lindane | <0.052 | <0.052 | 5.400 | <0.052 | <0.052 | <0.052 | <0.052 | <0.052 | <0.052 | <0.052 | <0.052 | 2.000 | 0.01 |
| Ɣ-Lindane | <0.059 | <0.059 | 2.140 | <0.059 | <0.059 | <0.059 | <0.059 | <0.059 | <0.059 | <0.059 | <0.059 | 2.000 | 0.01 |
| Total DDT | 4.300 | | 11.355 | | | | 0.915 | 0.657 | | | | 5.000 | 0.078 |
| p,p-DDT | <0.004 | <0.004 | 7.355 | <0.004 | <0.004 | <0.004 | 0.822 | 0.456 | <0.004 | <0.004 | <0.004 | 5.000 | 0.078 |
| o,p-DDT | 3.600 | <0.005 | 4.000 | <0.005 | <0.005 | <0.005 | <0.005 | <0.005 | <0.005 | <0.005 | <0.005 | 5.000 | 0.078 |
| p,p-DDE | <0.006 | <0.006 | <0.006 | <0.006 | <0.006 | <0.006 | <0.006 | <0.006 | <0.006 | <0.006 | <0.006 | 5.000 | 0.05 |
| o,p-DDE | <0.012 | <0.012 | <0.012 | <0.012 | <0.012 | <0.012 | <0.012 | <0.012 | <0.012 | <0.012 | <0.012 | 5.000 | 0.05 |
| p,p-DDD | <0.005 | <0.005 | <0.005 | <0.005 | <0.005 | <0.005 | 0.093 | 0.201 | <0.005 | <0.005 | <0.005 | 5.000 | 0.05 |
| op TDE | 0.700 | <0.006 | <0.006 | <0.006 | <0.006 | <0.006 | <0.006 | <0.006 | <0.006 | <0.006 | <0.006 | 5.000 | 0.05 |
| Fenthion | <0.071 | 5.184 | <0.071 | 0.2194 | 0.100 | 0.663 | <0.071 | 0.420 | 0.426 | 5489.24 | 14.998 | | |
| Diazinon | <0.001 | 1.300 | <0.001 | 1.301 | <0.001 | 0.653 | <0.001 | <0.001 | <0.001 | <0.001 | <0.001 | | |
| Permethrin | <0.015 | <0.015 | <0.015 | <0.015 | <0.015 | <0.015 | <0.015 | <0.015 | <0.015 | <0.015 | <0.015 | | |

TZ = Tanzania; Tanzania soil standard source: Environmental Management Regulations, 2007. Canada soil standard source: Soil, Ground Water and Sediment Standards for Use under Part XV.1 of the Environmental Protection Act, 2011.

**Table 4.** Persistent organic pesticide concentrations in selected profile samples at PPO Tengeru, Arusha, Tanzania. Values above the permissible levels are marked in red.

| Profile ID | Depth (m) | Pesticide Concentration (mg kg$^{-1}$) | | | | | | | |
|---|---|---|---|---|---|---|---|---|---|
| | | **p,p-DDT** | **o,p-DDT** | **p,p-DDE** | **o,p-DDE** | **p,p-DDD** | **o,p-DDD** | **p,p-TDE** | **Total DDT** |
| P1 | 0.5–1.0 | 0.540 | <0.005 | <0.012 | 0.151 | <0.005 | <0.005 | 2.026 | 2.717 |
| | 1.5–2.0 | 0.047 | <0.005 | <0.012 | <0.012 | <0.005 | <0.005 | 0.149 | 0.196 |
| | 2.5–3.0 | <0.004 | <0.005 | <0.012 | <0.012 | <0.005 | <0.005 | <0.006 | <DL |
| P2 | 0.5–1.0 | <0.004 | <0.005 | <0.012 | <0.012 | <0.005 | <0.005 | <0.006 | <DL |
| | 1.5–2.0 | <0.004 | <0.005 | <0.012 | <0.012 | <0.005 | <0.005 | <0.006 | <DL |
| | 2.5–3.0 | <0.004 | <0.005 | <0.012 | <0.012 | <0.005 | <0.005 | <0.006 | <DL |
| P3 | 0.5–1.0 | 0.639 | <0.005 | 0.284 | <0.012 | 0.743 | <0.005 | 2.843 | 4.509 |
| | 1.5–2.0 | <0.004 | <0.005 | <0.012 | <0.012 | 0.009 | <0.005 | <0.006 | <DL |
| | 2.5–3.0 | <0.004 | <0.005 | <0.012 | <0.012 | <0.005 | <0.005 | <0.006 | <DL |
| P4 | 0.5–1.0 | 0.639 | <0.005 | 0.315 | <0.012 | <0.005 | 0.691 | <0.006 | 1.645 |
| | 1.5–2.0 | <0.004 | <0.005 | 0.129 | <0.012 | <0.005 | 0.430 | <0.006 | 0.559 |
| | 2.5–3.0 | <0.004 | <0.005 | <0.012 | <0.012 | 0.110 | <0.005 | <0.006 | <DL |
| P5 | 0.5–1.0 | <0.004 | <0.005 | <0.012 | <0.012 | <0.005 | <0.005 | <0.006 | <DL |
| | 1.5–2.0 | <0.004 | <0.005 | <0.012 | <0.012 | <0.005 | 0.981 | <0.006 | 0.981 |
| | 2.5–3.0 | <0.004 | <0.005 | <0.012 | <0.012 | <0.005 | <0.005 | <0.006 | <DL |
| Permissible limits | Tanzania | 5.0 | 5.0 | 5.0 | 5.0 | 5.0 | 5.0 | 5.0 | 5.0 |
| | Canada | 0.078 | 0.078 | 0.078 | 0.05 | 0.05 | 0.05 | 0.05 | 0.078 |

The contamination of soils by DDT and lindane was more widespread at NHC Morogoro, with values of the total DDT ranging from 8.04 to 1591.3 mg kg$^{-1}$ in the surface soil samples (Table 4). Except for pp-DDT, all other metabolic forms of DDT were detected at quantities above the permissible levels (Tanzanian and Canadian permissible levels). The contamination of the surface soils by DDT was so widespread that every surface point sampled had detectable levels of DDT and its metabolites. Lindane was also detected at selected sampling points at the NHC Morogoro site, with detected total lindane values ranging from 13 to 181.02 mg kg$^{-1}$. The surface soil samples had detectable levels of other organochloride pesticides, such as Aldrin, diedrin and endosulfan (Table 5). Similarly, DDT and its metabolites were detected in each of the three profiles opened at NHC Morogoro, with the total DDT values ranging from 9.38 mg kg$^{-1}$ at a depth range of 1.5–2.0 m in profile no. 3 to 74.94 mg kg$^{-1}$ at a depth of 0.5–1.0 m in profile no. 2 (Table 6).

**Table 5.** Pesticide concentrations in surface soil samples at NHC Morogoro site, Arusha, Tanzania. Values above the permissible levels are marked in red.

| Field Sampling Point ID | A7 | B7 | C7 | D7 | A13 | B13 | C13 | D13 | A25 | B25 | C25 | D25 | TZ | Canada |
|---|---|---|---|---|---|---|---|---|---|---|---|---|---|---|
| **Pesticide** | Concentration (mg kg$^{-1}$ Soil) | | | | | | | | | | | | Permissible Level (mg kg$^{-1}$) | |
| Total Lindane | 13 | 30.11 | 181.02 | 203 | | | | | | | | | 2.0 | 0.01 |
| α-Lindane | ND | 12.01 | 87.11 | 119.50 | <0.005 | <0.005 | <0.005 | <0.005 | <0.005 | <0.005 | <0.005 | <0.005 | 2.00 | 0.01 |
| β-Lindane | 5.01 | 16.34 | 83.99 | 83.5 | <0.005 | <0.005 | <0.005 | <0.005 | <0.005 | <0.005 | <0.005 | <0.005 | 2.00 | 0.01 |
| ɣ-Lindane | 8.00 | 1.76 | 9.92 | ND | <0.005 | <0.005 | <0.005 | <0.005 | <0.005 | <0.005 | <0.005 | <0.005 | 2.00 | 0.01 |
| Total DDT | 1469.65 | 1591.3 | 319.89 | 889.11 | 8.04 | 313.27 | 1099.49 | 1526.12 | 17.112 | 33.54 | 37.03 | 418.16 | 5.00 | 0.078 |
| p,p-DDT | 8.99 | <0.004 | <0.004 | 5.24 | <0.004 | <0.004 | <0.004 | <0.004 | <0.004 | <0.004 | <0.004 | <0.004 | 5.00 | 0.078 |
| o,p-DDT | 901.89 | 822.05 | 161.11 | 462.6 | <0.005 | 115.55 | 557.52 | 873.79 | 1.06 | 10.35 | 15.25 | 60.43 | 5.00 | 0.078 |
| p,p-DDE | 236.41 | 425.97 | 88 | 39.42 | 4.71 | 114.29 | 299.95 | 286.19 | 6.35 | 18.98 | 10.22 | 229.08 | 5.00 | 0.05 |
| o,p-DDE | 12.98 | 23.29 | 6.79 | <0.0012 | <0.012 | <0.012 | 31.67 | 57.72 | 1.022 | <0.012 | 1.78 | 13.97 | 5.00 | 0.05 |
| p,p-DDD | 309.38 | 316.2 | 63.99 | 108.09 | 3.12 | 83.43 | 195.74 | 303.67 | 8.68 | 4.21 | 9.78 | 112.69 | 5.00 | 0.05 |
| op TDE | <0.005 | <0.005 | <0.005 | 273.76 | 0.21 | <0.005 | 14.61 | 4.75 | <0.005 | <0.005 | <0.005 | 1.99 | 5.00 | 0.05 |
| Aldrin | <0.003 | <0.003 | <0.003 | 17.01 | <0.003 | <0.003 | <0.003 | <0.003 | <0.003 | 0.511 | <0.003 | 1.3 | | |
| Diedrin | <0.003 | 16.63 | 47.51 | 5.65 | <0.003 | 146 | <0.003 | 106.07 | 26.4 | 8.65 | 1.4 | 110.47 | | |
| β-Endosulfan | <0.002 | 51.17 | 14.89 | 667.54 | <0.002 | <0.002 | <0.002 | 5.14 | <0.002 | 54.31 | <0.002 | 27.75 | | |
| α-Endosulfan | <0.001 | <0.001 | 8.45 | <0.001 | <0.001 | <0.001 | <0.001 | <0.001 | <0.001 | <0.001 | <0.001 | 48.96 | | |
| Simazine | <0.002 | <0.002 | <0.002 | <0.002 | <0.002 | <0.002 | <0.002 | <0.002 | <0.002 | <0.002 | <0.002 | <0.002 | | |

**Table 6.** Pesticide concentrations in selected profile samples at NHC Morogoro site.

| Profile ID | Depth (m) | Pesticide Concentration (mg kg$^{-1}$) | | | | | | |
|---|---|---|---|---|---|---|---|---|
| | | **p,p-DDT** | **o,p-DDT** | **p,p-DDE** | **o,p-DDE** | **p,p-DDD** | **o,p-DDD** | **Total DDT** |
| P1 | 0.5–1.0 | 1.00 | 10.06 | 0.36 | 3.53 | 1.922 | 10.35 | 27.22 |
| | 1.5–2.0 | 1.20 | 6.48 | 1.80 | 0.02 | 6.30 | 0.75 | 23.05 |
| P2 | 0.5–1.0 | 2.51 | 28.75 | 0.64 | 7.97 | 5.49 | 29.59 | 74.94 |
| | 1.5–2.0 | 1.38 | 5.33 | 0.79 | 10.96 | 1.03 | 5. 47 | 24.96 |
| P3 | 0.5–1.0 | 1.11 | 12.99 | 0.76 | 9.35 | 2.48 | 13.36 | 40.08 |
| | 1.5–2.0 | 0.31 | 3.63 | 0.11 | 0.89 | 0.69 | 3.74 | 9.38 |
| Permissible limits | Tanzania | 5.0 | 5.0 | 5.0 | 5.0 | 5.0 | 5.0 | 5.0 |
| | Canada | 0.078 | 0.078 | 0.078 | 0.05 | 0.05 | 0.05 | 0.078 |

At least one previous report indicated that the two current study sites were heavily contaminated by POPs, particularly DDT at NHC Morogoro and Lindane at PPO Tengeru [21]. Several other studies that documented the existence of POPs in the Tanzanian environment linked the underlying causes to mainly include discharges from agricultural pesticide use, malaria control programs and the disposal of obsolete stockpiles of POP-containing wastes [22–24]. In two separate studies, the NHC Morogoro and PPO Tengeru sites were ranked as the first and second priority sites for decontamination from POPs in reference to their suspected stock of POPs that were buried or unintentionally spilled over into the soils [19]. The observation in the current study that DDT is widespread and heavily present in most surface soil samples at NHC Morogoro than at PPO Tengeru corroborates assertions in a previous report by Niras [19]. In their report, Niras [19] identified NHC Morogoro and PPO Tengeru as the first and second priority POP hotspots and stated that the NHC site in Morogoro must be accorded the highest priority for any clean-up procedure. Ten years later, during this study, not much has changed, especially at NHC Morogoro, as the surface DDT levels are still high (1591 mg kg$^{-1}$).

Unlike in previous reports [19,21], we observed a decrease in the number of sampling points with detectable levels of lindane at the PPO Tengeru site compared to the NHC Morogoro site. Of the twelve sampling points at NHC Morogoro, four had detectable amounts of total lindane, compared to only one out of eleven sampling points at PPO Tengeru. Under natural open environmental conditions, lindane has a reported half-life in soil of about 708 days [25,26]. One reason for the purported reduction in dominance on

surface soils at PPO Tengeru could be its relatively good response (compared to DDT) to degradation by microbial agents [26] and thus the potential for native microbial communities' action on lindane at PPO Tengeru. Apart from DDT and lindane, sizable amounts of other organochloride pesticides were detected at NHC Morogoro but not PPO Tengeru. Accordingly, diedrin and β-endosulfan were more represented spatially on the surface soils of NHC Morogoro than aldrin or α-endosulfan, corroborating a previous observation that the NHC Morogoro site was indeed contaminated by multiple types of POPs [21].

### 3.2. Effects and Uptake of Residual DDT on Plant Growth under Screenhouse Conditions

Figures 1 and 2 present the effects of high (2308 mg kg$^{-1}$) or low (417 mg kg$^{-1}$) residual DDT concentrations in soils under screenhouse conditions. All test plants showed significantly ($p < 0.05$) higher DDT concentrations in high residual DDT soils than in low residual DDTs soils. Accordingly, calabash absorbed the highest amount of DDT per plant, and pumpkin absorbed the least amount of DDT while growing in high residual DDT soil. On the other hand, calabash accumulated the highest amount of DDT, while finger millet accumulated the least amount of DDT while growing in low residual DDT soil (Figures 1 and 2). All of the test plants showed reduced plant heights at high residual DDT compared to when growing in low residual DDT (Figure 1). The reduction in height was more pronounced in calabash and pumpkin than in the rest of the test plants (Figure 1).

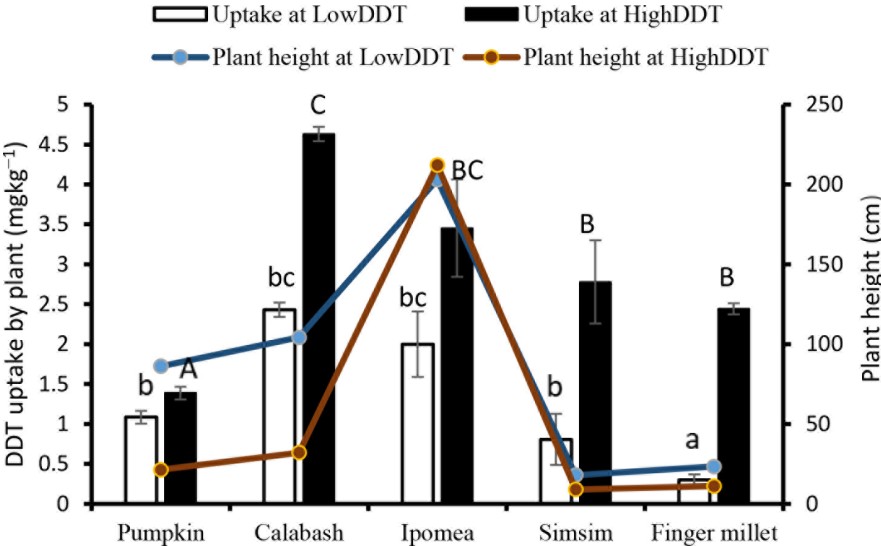

**Figure 1.** Effect of the soil's residual DDT amount (low or high) on plant height and uptake of DDT. Bars with the same type of letters are not significantly different at $p \leq 0.05$ according to Duncan's New Multiple Range Test. Error bars represent mean ± S.E., $n = 3$.

Similarly, sweet potato and calabash recorded the highest and second highest dry weight per plant while growing in high or low residual DDT (Figure 2). While finger millet had the lowest shoot dry weight per plant when grown in low residual DDT soils, pumpkin showed the lowest shoot dry weight when growing in high residual DDT soil. The test plants grown in low residual DDT soils had slightly higher shoot weight values than when grown in high residual DDT soils. The reductions in the dry shoot weights were more pronounced in calabash and pumpkin than in the rest of the test plants (Figure 2).

The DDT uptake by plants growing in potted soil has been shown to vary with the species and plant cultivars involved. In one study, for example, out of 23 different cultivars of castor oil plants, only 1 cultivar exhibited a significant ability to bioaccumulate DDT, which is attributable to its distinctly strong root system [26]. While the ability of calabash to absorb DDT from contaminated soils and bioaccumulate it in the roots and shoots has not been previously reported, pumpkin has been shown to have a high phytoremediation potential for DDT [27].

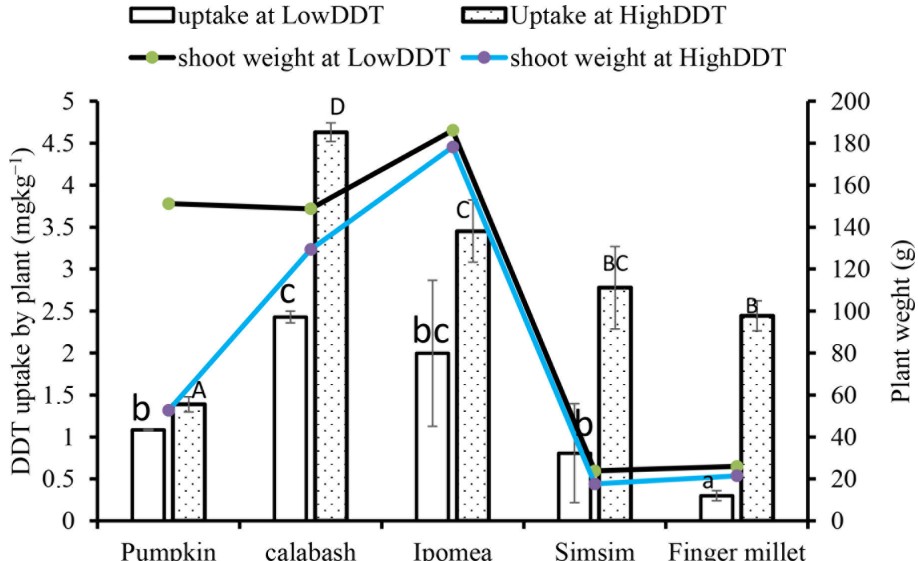

**Figure 2.** Effect of the soil's residual DDT amount (low or high) on uptake of DDT and shoot weight. Bars with the same type of letters are not significantly different at $p \leq 0.05$ according to Duncan's New Multiple Range Test. Error bars represent mean $\pm$ S.E., $n = 3$.

### 3.3. Plant Uptake of DDT and Metabolites from Contaminated Soils of PPO Tengeru and NHC Morogoro Sites

Figure 3 presents the data of two consecutive seasons on the plant uptake of DDT from contaminated soils under field conditions at the PPO Tengeru site. In the first 10 weeks of growing season 1, the above-ground biomass concentration of DDT was highest in sweet potato (98 ng g$^{-1}$), followed by pumpkin (61 ng g$^{-1}$), while in season 2, higher shoot concentrations of DDT were detected again in sweet potato (112 ng g$^{-1}$), followed by calabash (89 ng g$^{-1}$). The results show that sweet potato had a significantly ($p < 0.05$) higher concentration of DDT compared to the rest of the test crops used in the study. Overall, simsim exhibited the lowest performance in terms of the root and shoot concentrations of DDT under field conditions at PPO Tengeru (Figure 3).

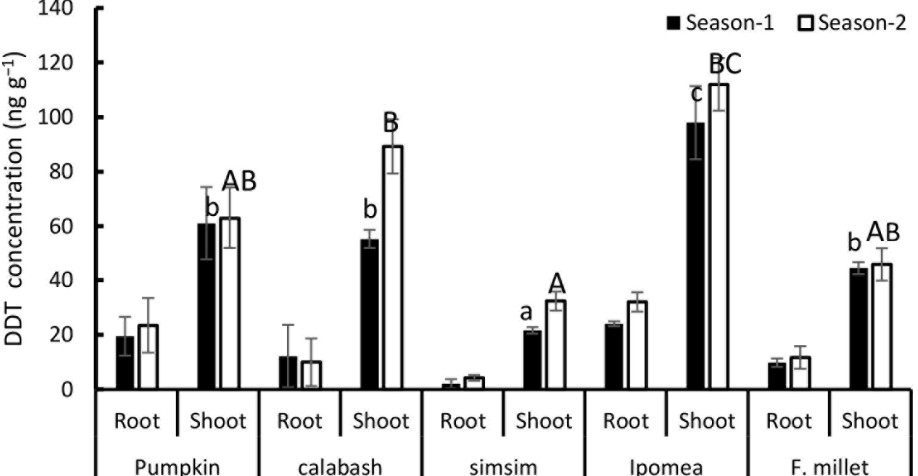

**Figure 3.** Mean DDT amounts accumulated in plant tissues 10 weeks after planting at PPO Tengeru contaminated site. Bars with the same type of letters are not significantly different at $p \leq 0.05$ according to Duncan's New Multiple Range Test. Error bars represent mean $\pm$ S.E., $n = 3$.

Along with calabash, sweet potato exhibited a consistently high ability to extract and bioaccumulate DDT from contaminated soils of NHC Morogoro and PPO Tengeru. A couple of studies have reported the potential for use of sweet potato in remediation of heavy metals and organochloride pesticide-contaminated soils [28–30]. Sweet potato has been reported as a hyperaccumulator plant with multiple metal and organochloride pesticide bioaccumulation capability [29]. We, in the present study, observed that sweet potato bioaccumulates DDT such that sizable proportions are stored in the below-ground and above-ground biomass, and that this is consistent with previous reports on the ability of various sweet potato varieties to bioaccumulate pesticides including heavy metals [28–32].

Figure 4 presents data on the plant uptake of DDT from contaminated soils for two 10-week growing seasons at the NHC Morogoro site. Accordingly, sweet potato, pumpkin and calabash exhibited the first, second and third highest total DDT absorption levels in season 1, while in season 2, sweet potato retained the highest spot in DDT absorption, followed by pumpkin and calabash, respectively.

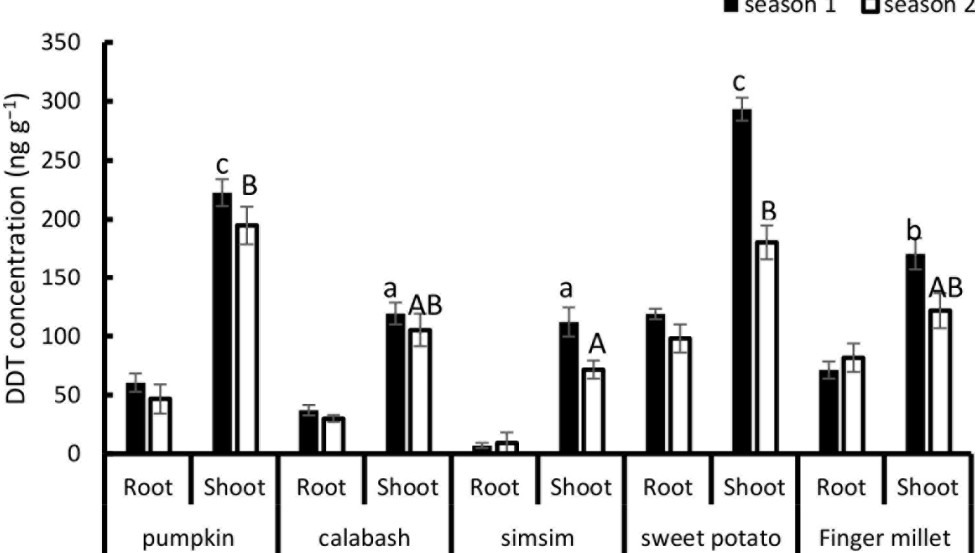

**Figure 4.** Mean DDT amounts accumulated in plant tissues 10 weeks after planting at NHC Morogoro contaminated site. Bars with the same type of letters are not significantly different at $p \leq 0.05$ according to Duncan's New Multiple Range Test. Error bars represent mean $\pm$ S.E., $n = 3$.

Unlike pumpkin, its wild relative in the *Cucurbitaceae* family—calabash—has not been sufficiently studied for its phytoremediation potential in POP-contaminated soils. We have shown in the present study that calabash can outperform its better-studied relative, pumpkin, in the pytoaccumulation of DDT, and thus, it presents a better alternate phytoremediation agent for DDT-contaminated soils. Accordingly, calabash has exhibited in almost all but one experiment that it can phytoaccumulate higher amounts of DDT in the roots and shoots compared to pumpkin.

Some authors have argued against the use of edible crop plants in any phytoremediation effort due to concerns on the possibility for the pollutants to move up the food chain and pose health risks to human beings [33,34]. However, research and the controlled use of edible crop species with good phytoaccumulation ability in the remediation of contaminated soils and sediments is justifiable if the phytobiomass generated in such undertakings is protected during vegetative growth, and upon harvest, properly destroyed and/or disposed.

Consistent with the observations made in the current study, a number of studies have reported testing and the use of edible crop species in the phytoremediation of heavy metals and persistent organic pesticides such as DDT [35–37]. Researchers, for example, reported the potential of sunflower along with three other edible crop species in the phytoremedi-

ation of endosulfan-polluted soils [35] and DDT [36]. Similarly, another study reported the potential for the use of various maize cultivars and alfalfa in the phytoremediation of DDT-contaminated soils [37]. Recently, the authors of [38] concluded that edible agricultural crops, namely oats, black beans, buckwheat and soybeans, were very effective as phytoremediation plants and could be used in the purification of any contaminated soil from heavy metals under the conditions of moderately dangerous pollution. Separately, Zea mays was reported as a strong candidate for the remediation of low to moderately copper-contaminated soils [39].

## 4. Conclusions

The success of any phytoremediation project is highly dependent on the selection of effective accumulator plants. We have shown in the current study that sweet potato and calabash, as well as pumpkin, can be deployed to help decontaminate NHC Morogoro and PPO Tengeru from DDT pollution. When an edible crop species such as sweet potato or pumpkin is used, the resulting phytobiomass should be protected against consumption or any other form of exposure to animals and human beings, as this may pose a health risks. Plants like calabash—which is neither used as human food nor consumed by livestock in most African countries, including Tanzania—should be given priority in future projects to decontaminate the NHC Morogoro and PPO Tengeru sites in Tanzania. Due to the huge amounts of DDT and its metabolites in the contaminated soil, especially at the NHC Morogoro site, remediation would require longer periods of time with repeated cycles of planting, removal and the proper disposal of bioaccumulator plants. Where possible, the intercropping of various competent bioaccumulator plants should be tested in future research.

**Author Contributions:** Conceptualization, experimental set up, data management and analysis and original draft write-up, H.J.T. Literature search, manuscript review and editing, B.R.S. All authors have read and agreed to the published version of the manuscript.

**Funding:** This study received no external funding.

**Institutional Review Board Statement:** Not applicable.

**Informed Consent Statement:** Not applicable.

**Data Availability Statement:** The data presented in this study are available upon request from the corresponding author.

**Conflicts of Interest:** The authors declare no conflicts of interest.

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
