# Peer review of "Potential of Calabash (Lagenaria siceraria) and Sweet Potato (Solanum tuberosum) for the Remediation of Dichlorodiphenyltrichloroethane-Contaminated Soils in Tanzania"

_soilsystems, doi:10.3390/soilsystems8010001_

Round 1

Reviewer 1 Report

Comments and Suggestions for Authors

See attached file.

Author Response

Response to Reviewers’ comments Manuscript ID: soilsystems-2735914

Reviewer 1

Starting with the title, I suggest that it be changed because it is important that phytoremediation is not carried out using edible plants

We have made more clarity around this matter by improving our explanation about handling of phytobiomass produced irrespective of whether the plant in question is edible or not. We have also shown current developments and publications towards using edible crop species in phytoremediation. Please see additional text in line 76-88 and line 373- 390.

It is important to show what will be done later with polluted materials (plants and soil)

Added text to show how polluted material were handled after the experiment (lines 220 -223) in the revised manuscript

It is necessary to carry out a broad characterization of the soil, to know the type of soil, texture and organic matter, pH, N,P,K

This information is now included in a tabular format. Please see Table 1 in the revised manuscript

Results for soil characterization not included

We have now shown these results in Table 1

Reviewer 2 Report

Comments and Suggestions for Authors

Point-by-point response to Comments and Suggestions for Authors

General comment: The topic of the manuscript is interesting and contribution is clear. However, there are some changes/modifications required.

·        The abstract is generally well-organized, but there could be clearer transitions between different sections, especially between the description of the experimental setup and the results obtained.

·        Consider adding a sentence summarizing the main objectives of the study at the beginning to provide a clear roadmap for readers.

·        Provide more details on the specific methodologies used in both the screen house and field experiments. For example, elaborate on the specific conditions of the screen house, including temperature, light, and watering protocols.

·        Clarify if there were any challenges or limitations encountered during the experiments that may have affected the results.

·        Justify the choice of the selected plant species and their relevance to the study. Are there specific characteristics of these plants that make them suitable for phytoremediation?

·        Explain the rationale behind the selection of DDT concentration levels (low and high) and how these levels relate to real-world contamination scenarios.

·        Discuss the implications of the three replications in terms of the reliability and robustness of the results.

·        In the Result and Discussion section; provide a more comprehensive discussion of the observed trends, especially regarding the differences in DDT concentration among plant species and in different soil conditions.

·        Address any unexpected or counterintuitive results and propose potential explanations or further investigations

·        The conclusion is concise but could be expanded to include practical implications of the findings. How might the observed plant-remediation relationships be applied in real-world environmental remediation efforts?

·        Suggest potential directions for future research based on the current findings.

·        Ensure consistency in units throughout the manuscript (e.g., mg kg-1, c.a.).

·                   

Comment 2:  Ensure consistency in the citation in the entire manuscript, E.g in the Introduction section, line no. 41 the citations are separated by a semicolon (;) as [1;2] but in line no. 53 the citation is separated by a comma (,) [7,8,9].

Comment 3: Regarding the punctuation and grammatical errors in the manuscript, do proofread and correct it. E.g Line no. 72 there is a comma (,) after sites and before the citation. “ contaminated sites, [14, 15].”.  Line no. 81 “ plant Protection office (PPO)” here plant must be capital as Plant Protection office (PPO). Line no 242 punctuation error, PPO Tengeru 241 [19].).

Comment 4: Ensure consistency in the font style and size in Table 2 and Table 3.

Comment 5: In Figures 5 and 6, include the X-axis name. Make all the figures with the same and standard resolution.

Comment 6: The English language needs substantial improvement.

Comments on the Quality of English Language

Please check the English grammar. Some minor corrections are required.

Author Response

Reviewer 2

General comment: The topic of the manuscript is interesting and contribution is clear. However, there are some changes/modifications required.

Thank you for the complement

The abstract is generally well-organized, but there could be clearer transitions between different sections, especially between the description of the experimental setup and the results obtained

Despite the word limit rule we have improved clarity and connectivity between paragraphs in the abstract

Consider adding a sentence summarizing the main objectives of the study at the beginning to provide a clear roadmap for readers

Added a description of the main objective at the end of the introduction section

Provide more details on the specific methodologies used in both the screen house and field experiments. For example, elaborate on the specific conditions of the screen house, including temperature, light, and watering protocols.

We have indicated the watering regime in the screenhouse, no specific controls of temperature and humidity were carried out during this experiment

Clarify if there were any challenges or limitations encountered during the experiments that may have affected the results.

No specific challenges were encountered

Justify the choice of the selected plant species and their relevance to the study. Are there specific characteristics of these plants that make them suitable for phytoremediation

An additional paragraph in lines 76-88 of the revised version shows the reasons for selection of test plants in this study

Explain the rationale behind the selection of DDT concentration levels (low and high) and how these levels relate to real-world contamination scenarios

Added explanation on choice of DDT levels lines 173-175 of the revised manuscript

Discuss the implications of the three replications in terms of the reliability and robustness of the results

We did not include this in the text as it is a standard statistical practice to replicate agric/biological experiments at least 3 times or more depending on the budgetary flexibility.

In the Result and Discussion section; provide a more comprehensive discussion of the observed trends, especially regarding the differences in DDT concentration among plant species and in different soil conditions

We have improved the results and discussion section

Address any unexpected or counterintuitive results and propose potential explanations or further investigations

We did not find any unexpected results that needed explanation

The conclusion is concise but could be expanded to include practical implications of the findings. How might the observed plant-remediation relationships be applied in real-world environmental remediation efforts?

Conclusion improved

Suggest potential directions for future research based on the current findings.

Done at the conclusion section

Ensure consistency in units throughout the manuscript (e.g., mg kg-1, c.a.).

Units checked, throughout

Comment 2: Ensure consistency in the citation in the entire manuscript, E.g in the Introduction section, line no. 41 the citations are separated by a semicolon (;) as [1;2] but in line no. 53 the citation is separated by a comma (,) [7,8,9].

Citations checked and corrected

Comment 3: Regarding the punctuation and grammatical errors in the manuscript, do proofread and correct it. E.g. Line no. 72 there is a comma (,) after sites and before the citation. “ contaminated sites, [14, 15].”. Line no. 81 “ plant Protection office (PPO)” here plant must be capital as Plant Protection office (PPO). Line no 242 punctuation error, PPO Tengeru 241 [19].).

Checked and corrected

Comment 4: Ensure consistency in the font style and size in Table 2 and Table 3.

Corrected for all tables

Comment 5: In Figures 5 and 6, include the X-axis name. Make all the figures with the same and standard resolution.

All Figures (Figure 1 to 4) have been reworked and X-axis names standardised

Comment 6: The English language needs substantial improvement

English has been improved

Please check the English grammar. Some minor corrections are required.

Worked and corrected grammatical errors throughout

Reviewer 3 Report

Comments and Suggestions for Authors

My opinion is that the manuscript is well written, and the results of well designed experiment are analyzed and presented in an appropriate manner. Therefore, I think that this manuscript should be accepted with minor revision. My minor comments are below.

Abstract: (and throughout the manuscript) there is a dot after number somewhere, and somewhere a space is missing.

Even though „were replicated thrice“ is correct; „were done in triplicate“ is usually used.

Why ca. for DDT concentration? (and throughout the text?)

L50-51 Change to „…by either: a) non-combustion…or, b) combustion (remove „through“ twice

Table 2 – Explanation why the text is in red should be in table title, not below

Table 3  - the same as Table 2

Figure 1: Typing error „height“

L320 potatto is in italic

In Figures 3 and 4 it would be good to place letters above columns

In most figures instead of „Error bars represent standard errors of the mean from three replications)“, you could write (mean ± S.E., n=3)

Author Response

Reviewer 3

My opinion is that the manuscript is well written, and the results of well designed experiment are analyzed and presented in an appropriate manner. Therefore, I think that this manuscript should be accepted with minor revision.

Thank you for the comment, minor revisions have been done

Abstract: (and throughout the manuscript) there is a dot after number somewhere, and somewhere a space is missing

Corrected

Even though „were replicated thrice“ is correct; „were done in triplicate“ is usually used.

Corrected as suggested

Why ca. for DDT concentration? (and throughout the text?)

Corrected by removing ca in the text and in abstract

L50-51 Change to „…by either: a) non-combustion…or, b) combustion (remove „through“ twice

Corrected as suggested

Table 2 – Explanation why the text is in red should be in table title, not below

Corrected

Table 3 - the same as Table 2

Corrected

Figure 1: Typing error „height

corrected

L320 potatto is in italic

corrected

In Figures 3 and 4 it would be good to place letters above columns

Corrected for all figure

In most figures instead of „Error bars represent standard errors of the mean from three replications)“, you could write (mean ± S.E., n=3)

Adopted as suggested

Reviewer 4 Report

Comments and Suggestions for Authors

Dear Editor and Authors,

I have carefully read your manuscript titled: Potential of Calabash (Lagenaria siceraria) and Sweet Potato (Solanum tuberosum) for Remediation of Ddt-Contaminated Soils in Tanzania. The paper is interesting and valuable. The figures are attractive, and the tables are full of very interesting results.

Calabash (Lagenaria siceraria) and sweet potato (Solanum tuberosum) are not typically known as hyperaccumulators, but some plants have shown the ability to enhance the degradation or absorption of certain pollutants in the soil.  I have doubts about whether the research topic is named appropriately because I am not sure whether it is possible to talk about the potential for phytoremediation of edible plants, or perhaps in their case it is more appropriate to describe their ability to accumulate contaminants in order to maintain food safety. However, this is not a resolved issue, so please do not treat my comment as criticism, but rather as a starting point for discussion. Understanding how different plant species, including edible ones, respond to and mitigate soil contaminants contributes to overall knowledge about ecosystem health and is crucial. Perhaps we should only be careful in drawing conclusions and calling them phytoremediation potential.

Please see my minor comments below:

- In my opinion, it is worth adding a graphical abstract to this manuscript, it will definitely encourage potential readers to read the text;

- line 12 - dichlorodiphenyltrichloroethane - there is no need to start the name of an organic compound with a capital letter in the middle of a sentence;

- line 16 - unnecessary full stop;

- line 17 - 2308 mg - missing space;

- please use different keywords than words in the title;

- please check the temperature units in the whole text;

- please check style of the text in the text and in the tables (superscripts, spaces, text alignment etc.);

- please add full information about analytical equipment (producer, country, model).

Author Response

Reviewer 4

S/N

Reviewer’s comment/suggestions

Response/action taken

1

Calabash (Lagenaria siceraria) and sweet potato (Solanum tuberosum) are not typically known as hyperaccumulators, but some plants have shown the ability to enhance the degradation or absorption of certain pollutants in the soil.

 I have doubts about whether the research topic is named appropriately because I am not sure whether it is possible to talk about the potential for phytoremediation of edible plants, or perhaps in their case it is more appropriate to describe their ability to accumulate contaminants in order to maintain food safety.

However, this is not a resolved issue, so please do not treat my comment as criticism, but rather as a starting point for discussion.

Understanding how different plant species, including edible ones, respond to and mitigate soil contaminants contributes to overall knowledge about ecosystem health and is crucial. Perhaps we should only be careful in drawing conclusions and calling them phytoremediation potential.

We have revised the document text to emphasize that irrespective of whether the test crop is edible or not the phytobiomass produced must be protected prior to proper disposal to avoid risk of pollutants entering the food chain and moving up the trophic levels

Thank you for the comment, Yes  we have revised the document by adding information showing scientific works through which edible plants have been successfully tested and, in some cases, deployed to help decontaminate polluted soils (Please see in the revised version line 76 to 88 and again lines 371 to 389

In my opinion, it is worth adding a graphical abstract to this manuscript, it will definitely encourage potential readers to read the text;

Yes, we have submitted a graphical abstract for inclusion in the revised version of the manuscript

- line 12 - dichlorodiphenyltrichloroethane - there is no need to start the name of an organic compound with a capital letter in the middle of a sentence;

Corrected

- line 16 - unnecessary full stop; - line 17 - 2308 mg - missing space;

corrected

- please use different keywords than words in the title; -

corrected

please check the temperature units in the whole text;

corrected

- please check style of the text in the text and in the tables (superscripts, spaces, text alignment etc.);

Checked and corrected

- please add full information about analytical equipment (producer, country, model)

Information  added

Round 2

Reviewer 1 Report

Comments and Suggestions for Authors

Dear authors,

Thank you for responding to the suggestions, there was a substantial improvement in your manuscript. For my part, I approve.

Congratulations.